

# Comparative analysis of codon usage patterns and phylogenetic implications of five mitochondrial genomes of the genus *Japanagallia* Ishihara, 1955 (Hemiptera, Cicadellidae, Megophthalminae)

Min Li[1], Jiajia Wang[2], Renhuai Dai[1], Guy Smagghe[1,3,4], Xianyi Wang[5] and Siying You[1]

[1] Institute of Entomology, Guizhou University, The Provincial Key Laboratory for Agricultural Pest Management Mountainous Region, Guiyang, Guizhou, China
[2] College of Biology and Food Engineering, Chuzhou University, Chuzhou, Anhui, China
[3] Cellular and Molecular Life Sciences, Department of Biology, Brussels, Belgium
[4] Laboratory of Agrozoology, Dep. of Crop Protection, Ghent University, Ghent, Belgium
[5] Engineering Research Center of Medical Biotechnology, School of Biology and Engineering, Guizhou Medical University, Guiyang, Guizhou, China

Corresponding author
Renhuai Dai, dmolbio@126.com

## ABSTRACT

*Japanagallia* is a genus of Cicadomorpha in the family of leafhoppers that are plant piercing-sucking insects, and it is difficult to distinguish by morphological characteristics. So far, only one complete mitochondrial genome data has been reported for the genus *Japanagallia*. Therefore, in order to better understand this group, we assembled and annotated the complete mitochondrial genomes of five *Japanagallia* species, and analyzed their codon usage patterns. Nucleotide composition analysis showed that AT content was higher than GC content, and the protein-coding sequences preferred to end with A/T at the third codon position. Relative synonymous codon usage analysis revealed most over-represented codon ends with A or T. Parity plot analysis revealed the codon usage bias of mitochondrial genes was influenced by both natural selection and mutation pressure. In the neutrality plot, the slopes of regression lines were < 0.5, suggesting that natural selection was playing a major role while mutation pressure was of minor importance. The effective number of codons showed that the codon usage bias between genes and genomes was low. Correspondence analysis revealed that the codon usage pattern differed among 13 protein-coding genes. Phylogenetic analyses based on three datasets using two methods (maximum likelihood and Bayesian inference), restored the Megophthalminae monophyly with high support values (bootstrap support values (BS) = 100, Bayesian posterior probability (PP) = 1). In the obtained topology, the seven *Japanagallia* species were clustered into a monophyletic group and formed a sister group with *Durgade*. In conclusion, our study can provide a reference for the future research on organism evolution, identification and phylogeny relationships of *Japanagallia* species.

## INTRODUCTION

Codon usage bias (CUB) refers to the fact that synonymous codons in most organisms are not uniformly used in the coding DNA sequence, but preferentially use some specific codons (*Behura & Severson, 2012*). This property is unique to specific species and may vary in the genes of the same organism (*Wei et al., 2014*). Codon usage pattern is affected by many factors such as nucleotide composition, mutation pressure, natural selection, protein hydrophobicity (*Das, Paul & Dutta, 2006*; *Yadav & Swati, 2012*; *Zhao & Liang, 2016*). At the same time, studies on the use of different biological codons have shown that CUB is associated with gene expression, structure and function, translation elongation rate, and molecular mechanisms of translation of any protein (*Zhang, Dai & Dai, 2013*; *Yu et al., 2015*). Thus, CUB analysis helps to further improve the level of protein expression and to understand the genomic structure and evolutionary features of organisms (*Galtier et al., 2018*).

Megophthalminae (Hemiptera, Auchenorrhyncha, Cicadellidae) is one of the 27 leafhoppers subfamilies, which consists of 79 genera, four tribes and 742 species that have a wide distribution in the word (https://www.catalogueoflife.org/data/taxon/9DMXQ) (accessed on 24 March 2023). Megophthalminae leafhopper mostly feeds on Poaceae, Leguminosaceae (*Viraktamath, 2011*), and some species are agricultural pests, such as *Anaceratagallia venosa* (*Fourcroy, 1785*); this species causes direct feeding damage and indirect damage to plants by absorbing plant juice and spreading plant pathogens (*Viraktamath, 2011*; *Wilson & Turner, 2010*).

In the past decades, the mitochondrial genome (mitogenome) has been widely used as an important molecular data for molecular evolution, population genetics, phylogenetics and species identification of insects (*Yu & Zhang, 2021*; *Du et al., 2021*; *Tian et al., 2022*). Taxonomy researchers have also applied mitochondrial genome to the phylogenetic studies of Cicadellinae. In Megophthalminae, the first and second complete mitochondrial genomes (*Japanagallia spinosa* and *Durgade nigropicta*) were reported by *Wang et al. (2017)*, which verified the monophyly of this subfamily. To date, only seven complete mitochondrial genomes have been reported, including the five newly generated in this study.

At present, there are only one complete mitogenome of the genus *Japanagallia* has been sequenced and reported (*J. spinosa*, GenBank: NC_035685) (*Wang et al., 2017*), which greatly limits our understanding of the phylogenetic relationship among members of this genus and is not sufficient for phylogenetic analysis of Megophthalminae. *Wang et al. (2017)* reported two mitogenomes of Megophthalminae and analyzed their phylogenetic relationships, and the results showed that *Japanagallia* is monophyletic, *Japanagallia* and *Durgade* are sister groups. But due to having only one species, the taxonomic status of the genus cannot be well confirmed. At the same time, the species of the genus *Japanagallia* have similar morphological characteristics, making species identification a very difficult and challenging task. *Viraktamath, Dai & Zhang (2012)* misidentified *Japanagallia hamata* as *Japanagallia neohamata* due to its very similar color and form (*Li, Dai & Li, 2014*).
Therefore, molecular techniques are needed to help us more accurately identify the species and understand their phylogenetic relationships.

In this study, we sequenced and annotated the new complete mitogenome of five species of *Japanagallia* (*Japanagallia curvipenis*, *Japanagallia malaisei*, *Japanagallia multispina*, *Japanagallia turriformis* and *Japanagallia* sp.), increased the molecular data of Cicadellidae, and conducted comparative mitogenomic analyses from the aspects of genome size, nucleotide composition and codon usage bias. Furthermore, we combined our data with the previously published available mitogenomes of 73 Cicadellidae insects from 13 subfamilies (Table S1), and reconstructed the phylogenetic relationships between the five newly sequenced species and other mitochondrial genomes of Cicadellidae, based on 13 protein-coding genes and two ribosomal RNA genes concatenated nucleotide sequences. It provides novel and in-depth insights for further investigations into mitogenomic characteristics, codon usage patterns, the biological evolution of *Japanagallia*, and the phylogeny of Megophthalminae species.

## MATERIALS AND METHODS

### Sample preparation and DNA extraction

Collection information for the specimens used in this study was provided in Table S2. Live specimens were collected by sweep nets and immediately preserved in 100% ethanol and stored at −20 °C until identification and DNA extraction. Samples were identified by their morphological characteristics (*Rakitov, 1998*; *Dietrich, 2005*; *Viraktamath, 2011*). Total genomic DNA was extracted from the whole body of adult males using a DNeasy©Tissue Kit (Qiagen, Hilden, Germany) following the manufacturer's protocols. Genomic DNA were stored at −20 °C, and male external genitalia were preserved in glycerol at room temperature. Voucher adult specimens with male genitalia and DNA sample have been deposited at the Institute of Entomology, Guizhou University, Guiyang, China (GUGC).

### Sequence assembly and annotation

Genomes of the five species were sequenced by next-generation sequencing using Illumina HiSeq 4000 (Berry Genomic, Beijing, China). Each mitogenome was assembled using Geneious Primer (v.2019.2.1) (*Kearse et al., 2012*) using *J. spinosa* (GenBank NC_035685) (*Wang et al., 2017*) as the reference.

The assembled mitogenomes were annotated using MITOS web server with the invertebrate genetic code (*Bernt et al., 2013*), compared with mitogenomes of other leafhoppers, and identified by BLAST searches on NCBI (https://blast.ncbi.nlm.nih.gov/Blast.cgi) to confirm that the sequences were correct (*Altschul et al., 1997*). The 13 protein-coding genes (PCGs) boundaries were identified by ORF Finder of Geneious Prime applying the invertebrate mitochondrial genetic code. Abnormal start and stop codons were determined through comparisons with other insects. The locations and secondary structures of 22 tRNA genes were estimated using tRNAscan-SE version 1.21 (http://lowelab.ucsc.edu/tRNAscan-SE/) (*Lowe & Eddy, 1997*) and ARWEN version 1.2 (*Laslett & Canbäck, 2008*). The location of *16S* rRNA was determined according to the

location of *trnL2* and *trnV*. The target gene was compared with the existing mitochondrial genome sequence by BLAST to find out the location of *12S* rRNA and control region.

## Mitochondrial genomic composition

The compositional properties of the 13 protein-coding genes in the mitochondrial genome of five *Japanagallia* species, including all nucleotides composition (A, T, G and C), nucleotides composition at the third position of codons (A3, C3, T3 and G3), all GC and AT contents, the average of nucleotide G and C present at first and second positions of codon (GC12), GC contents at the first, second, third position (GC1, GC2 and GC3) in percentages, were calculated by MEGA6.06 software (*Tamura et al., 2013*). Meanwhile, in order to understand the bias of nucleotide, we calculated skew value (GC and AT) (*Tillier & Collins, 2000*) using the following formula:

$$\text{GC skew} = \frac{\text{G} - \text{C}}{\text{G} + \text{C}} \text{ and AT skew} = \frac{\text{A} - \text{T}}{\text{A} + \text{T}}.$$

## Relative synonymous codon usage

The relative synonymous codon usage (RSCU) definition refers to the relative probability between synonymous codons encoding corresponding amino acids for a specific codon (*Sharp, Tuohy & Mosurski, 1986*). If the value of RSCU > 1.6, the codon is over-represented, and if RSCU < 0.6, the codon is under-represented (*Wong et al., 2010*). When the RSCU value greater than 1 or less than 1 represent two types of codons, namely high bias (>1) and low bias (<1). However, when RSCU = 1, we consider that the codon usage lacks bias (*Sharp & Li, 1986*). The RSCU values of each codon of 13 protein-coding genes in the mitochondrial genome were calculated in the software MEGA6.06 (*Tamura et al., 2013*). Based on the relative synonymous codon values of 13 protein-coding genes of five newly sequenced mitogenomes, a heat map was drawn through TBtools v1.120 (*Chen et al., 2020*)

## Neutrality plot

To understand the effects of mutational pressure and natural selection on codon usage bias in species of five *Japanagallia* species, the neutrality plot can be drawn with GC12 as the ordinate and GC3 (average of GC1 and GC2) as the abscissa. In the plot, as the slope gets closer to or equal to one, it indicates that mutational pressure plays a decisive role in codon usage. Whereas when the slope is far from one, it means that codon use is mainly influenced by other forces, such as natural selection (*Sueoka, 1988*; *Sueoka, 1999*).

## Parity rule two plot

The parity rule two (PR2) plot is drawn with AT-bias value [A3/(A3 + T3)] as the *y*-axis and GC-bias value [G3/(G3 + C3)] as the *x*-axis to analyze the role of natural selection and mutational pressure on codon selection (*Sueoka, 1995*). The central point 0.5 of the graph shows that the influence of natural selection and mutational pressure are equal (*Yang et al., 2015*).

## Effective number of codons

The effective number of codons (ENC) is often widely used to evaluate whether synonymous codons are used equally, with values ranging from 20 to 61 (*Wright, 1990*). A lower ENC value means that higher codon usage bias, and vice versa. Typically, an ENC value 35

reflects a strong preference for codons (*Wright, 1990*; *Jiang et al., 2008*). The ENC value 20 indicates that each amino acid is encoded with only one codon, while the ENC value 61 indicates that each codon can be used equally when encoding amino acids. The ENC values were estimated in MEGA6.06 (*Tamura et al., 2013*).

## Correspondence analysis

The correspondence analysis (COA) is a multivariate statistical technique used to determine the major trends of codon usage variation among genes (*Clarke & Greenacre, 1985*; *Greenacre, 1894*; *James & McCulloch, 1990*). To understand trends in codon usage in axis1 and aix2 (*Perrière & Thioulouse, 2002*; *Shields & Sharp, 1987*), we performed COA analysis based on the RSCU values for 13 protein-coding genes in the mitogenomic using Pest software.

## Grand average of hydropathy

The grand average of hydropathy (GRAVY) score was calculated by the sum of the products of the frequency of each amino acid and the corresponding hydropathy index of each amino acid (*Kyte & Doolittle, 1982*). The positive GRAVY value indicates that the protein is essentially hydrophobic, while the negative value represents that the protein is hydrophilic in nature. GRAVY is calculated using Galaxy (*Afgan et al., 2018*).

## Phylogenetic analysis

The phylogenetic analysis was performed using whole mitogenomes of the 81 leafhopper species and five treehopper species as the ingroup, *Cervaphis quercus* (NC_024926) and *Cacopsylla coccinea* (NC_027087) were selected as members of the outgroups. Taxonomic information and mitochondrial accession numbers for each species are listed in Table S2. Three datasets were concatenated for phylogenetic analysis: (1) amino acid sequences of 13 PCGs with 3,594 amino acids (AA); (2) 13 PCGs and 2 rRNA genes (*12S* and *16S* rRNA) with 12,281 nucleotides (PCG-rRNA); (3) the first and second codon positions of the 13 PCGs and 2 rRNA genes (*12S* and *16S* rRNA) with 8,686 nucleotides (PCG12-rRNA). Each of the PCG (without stop codons) was aligned individually based on the invertebrate mitochondrial genetic code using the MAFFT algorithm in the Translator X online server (http://translatorx.co.uk/) (*Abascal, Zardoya & Telford, 2010*). Two rRNA genes were aligned using MAFFT v7 (https://mafft.cbrc.jp/alignment/server/index.html) with the Q-INS-I strategy (*Katoh, Rozewicki & Yamada, 2019*), and the poorly aligned positions and divergent regions were removed using the Gblocks 0.91b component in PhyloSuite v1.2.2 (*Talavera & Castresana, 2007*; *Zhang et al., 2020*). Next, the clipped sequences were concatenated into three different datasets using MEGA 6.06 (*Tamura et al., 2013*), the position of each gene fragment was recorded, and the resulting 15 alignments were manually checked and corrected.

The phylogenetic analyses were reconstructed using Bayesian Inference (BI) and Maximum Likelihood (ML) methods. The three concatenated datasets obtained were converted into "nex" and "phy" formats using Mesquite v2.75. The optimal partition scheme for each dataset and the best model for each partition was determined in PartitionFinder version 2.1.1 (*Miller, Pfeiffer & Schwartz, 2010*) using the Akaike

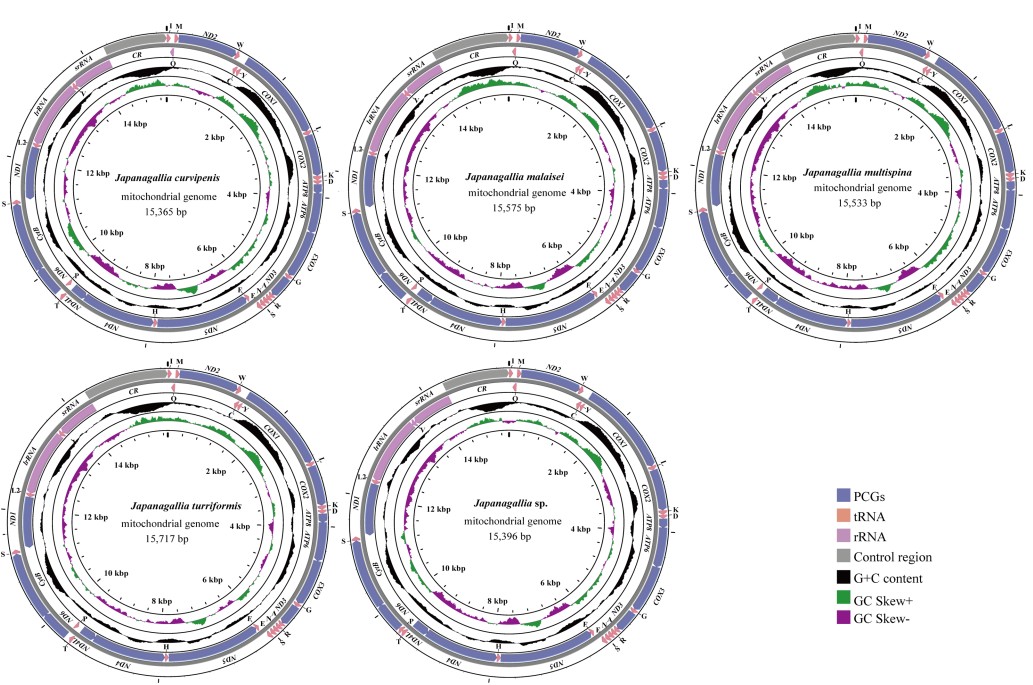

**Figure 1** **The circular map of the complete mitochondrial genome of five *Japanagallia* species.**

information criterion correct (AICc), and using the "greedy" algorithm with branch lengths estimated an "linked" a greedy search algorithm (*Lanfear et al., 2017*). Maximum likelihood (ML) phylogenetic trees were constructed with IQ-TREE v1.6.3 (*Nguyen et al., 2015*) using an ultrafast bootstrap (UFB) approximation approach for 10,000 replicates (*Nguyen et al., 2015*). Bayesian analysis used defaulted settings by simulating four independent runs for 100 million generations and sampling once every 1,000 generations in MrBayes 3.2.6 (*Huelsenbeck & Ronquist, 2001*; *Nylander et al., 2004*), after the average standard deviation of split frequencies fell below 0.01, the initial 25% of samples were discarded as burn-in, and the remaining trees were used to generate a consensus tree and calculate the posterior probability (PP) of each branch. The beautification of phylogenetic trees using FigTree v1.4.4 and Adobe Illustrator CS6 software.

# RESULTS

## Mitogenome organization and composition analysis

The size of the complete mitochondrial genomes of the five *Japanagallia* species were 15,365 bp in *J. curvipenis*, 15,575 bp in *J. malaise*, 15,533 bp in *J. multispina*, 15,717 bp in *J. turriformis*, and 15,396 bp in *Japanagallia* sp., respectively. Figure 1 shows the circular maps of five *Japanagallia* species mitogenomes. Like the mitochondrial genome of most leafhoppers, each of the newly sequenced mitogenomes contained 37 coding genes (13 PCGs, 2 rRNAs genes, 22 tRNAs genes), and a control region.

At the same time, in order to accurately understand the codon bias, we analyzed the 13 protein-coding genes of each genome of five *Japanagallia* species. The nucleotide base

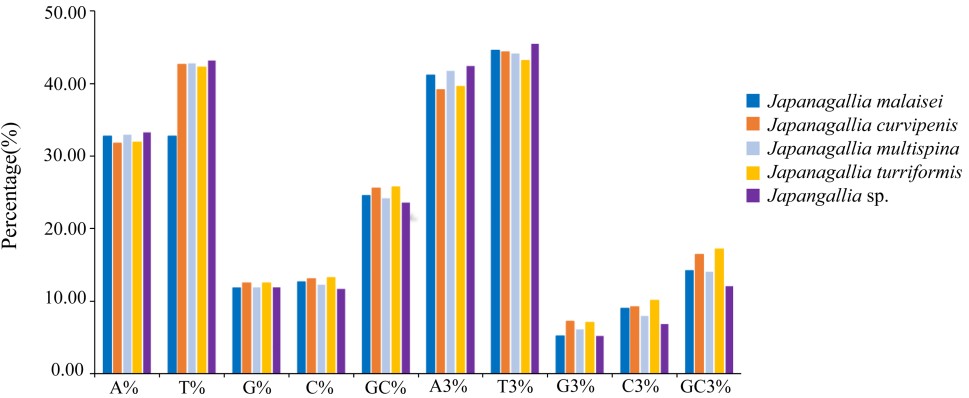

**Figure 2** Overall nucleotide composition and its 3rd codon position.

frequencies (A, T, G, C and GC) as well as with frequencies at its third codon position (A3, T3, G3 and C3) were calculated (Table S3). Figure 2 depicted the average nucleotide compositions of all the genes. We found that the average percentage of nucleotide T was the most abundant base followed by nucleotide A, C and G, the trend of the nucleotide frequencies was T>A>C>G. Likewise, the average percentage of nucleotide composition at the third codon position revealed that the average percentage of T3 was the highest followed by A3, C3 and G3 (T3>A3>C3>G3).

Moreover, the average percentage of GC contents at different codon positions were also analyzed (Fig. 3), and the results showed that the percentage of GC2 content was higher than the content GC1 and GC3, and the content GC3 was the lowest in the five *Japanagallia* species, the overall percentage of GC content were less than 50%. Therefore, this compositional study suggested that the A, T, G, and C nucleotides are unequally distributed, and the genes mostly preferred T or A nucleotide, while the C or G were less preferred.

### Relative synonymous codon usage analysis

We analyzed the RSCU value of each codon corresponding to amino acid in the CDS of 13 mitochondrial genes from the five *Japanagallia* species. By analyzing the overall RSCU (Table 1) of the CDS of mitochondrial 13 protein-coding genes, we observed that T ending codons were mostly preferred followed by A ending ones and thus the frequency of A or T was higher than that of G and C ending codons (Table S4). At the same time, we performed hierarchical clustering of the heatmap (Fig. 4). The color and degree of intensity clearly indicated the RSCU value and it varies from blue (low value of RSCU) to red (high value of RSCU). Orange and red color indicate over-represented codons (RSCU > 1.6), while blue under-represented codons (RSCU < 0.6). Figure 4 and Table S4 show that the codons UUA encoding Leucine amino acid (RSCU > 3) was favored by nature in all the 13 mitochondrial genes of five *Japanagallia* species.

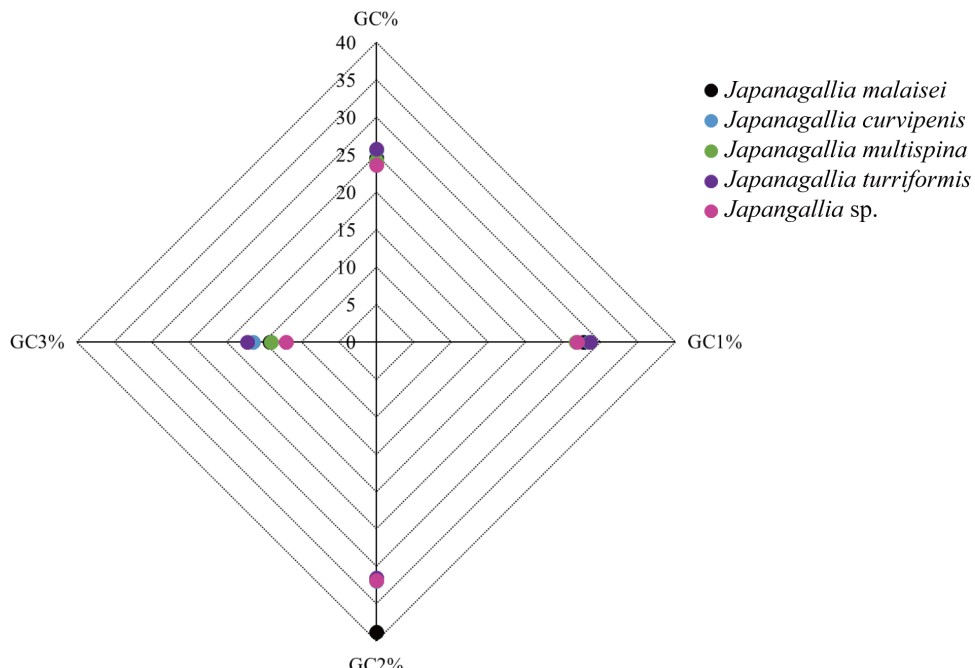

**Figure 3** Comparative analysis of GC contents in five *Japanagallia* species for 13 mitochondrial protein-coding genes.

**Table 1** Comparative analysis of synonymous codon usage in five *Japanagallia* species for 13 mitochondrial protein-coding genes.

| Sl. No. | Genes | Total Over-represented Codons RSCU > 1.6 | RSCU > 1.0 | Total preferred codons |
|---------|-------|------------------------------------------|------------|------------------------|
| 1 | ATP6 | 15 | 13 | 28 |
| 2 | ATP8 | 16 | 4 | 20 |
| 3 | COX1 | 17 | 11 | 28 |
| 4 | COX2 | 15 | 13 | 28 |
| 5 | COX3 | 13 | 15 | 28 |
| 6 | CYTB | 14 | 14 | 28 |
| 7 | ND1 | 16 | 8 | 24 |
| 8 | ND2 | 18 | 10 | 28 |
| 9 | ND3 | 14 | 12 | 26 |
| 10 | ND4 | 16 | 7 | 23 |
| 11 | ND4L | 17 | 5 | 22 |
| 12 | ND5 | 15 | 12 | 27 |
| 13 | ND6 | 18 | 8 | 26 |

## Neutrality plot analysis

In our analysis, the neutrality plot results showed that the regression coefficients of GC12 *versus* GC3 in all 13 mitochondrial genes were less than 0.5 (*ATP6*: 0.076, *ATP8*: 0.017, *COX1*: 0.073, *COX2*: 0.211, *COX3*: −0.063, *CYTB*: 0.114, *ND1*: 0.235, *ND2*: 0.181, *ND3*:

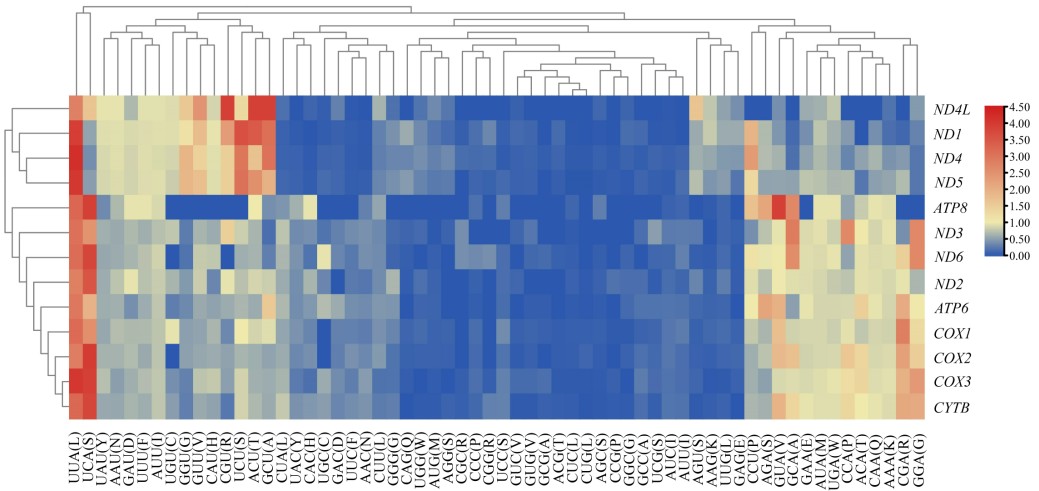

**Figure 4** **Clustering heatmap showing 13 mitochondrial protein-coding genes using relative synonymous codon usage (RSCU) values.** The $x$-axis and $y$-axis indicate the hierarchical clustering of codon frequencies and 13 protein-coding genes, respectively.

0.032, *ND4*: 0.025, *ND4L*: 0.052, *ND5*: 0.040, *ND6*: 0.231) (Fig. 5), indicating that mutation pressure effect accounted only 1.69%–23.46%. Therefore, the above results suggested that natural selection might have played a dominant role, whereas mutation pressure and other factors might have played a lesser role in shaping CUB (*Uddin & Chakraborty, 2020*).

## Parity plot analysis

The overall GC bias [G3/(G3 + C3)] and AT bias [A3/(A3 + T3)] was 0.441 and 0.514, separately. We plotted [G3/(G3 + C3)] along the $x$-axis and [A3/(A3 + T3)] along the $y$-axis of the graphical plot for the 13 mitochondrial genes of five *Japanagallia* species (Fig. 6). According to the parity plot, the central coordinate (0.5, 0.5) indicates that mutation pressure is the only force of codon bias, that is, the frequencies of nucleotides A and T are equal to the frequencies of C and G at the third position (*Kawabe & Miyashita, 2003*). In contrast, the degree of deviation from the center that is both mutations and selection will be involved in codon usage bias, with G and C unequal in the use of A and T bases (*Sueoka, 1995*). In our present study, overall the values of GC bias < 0.5 and AT bias > 0.5 (Table 2) can predict the preference of pyrimidine (U/C) over purine (A/C) at the third codon position (*Zhang et al., 2018*). Furthermore, we observed an uneven distribution of genes in the four regions of the PR2 plane, confirming the role of mutational pressure and natural selection in the use of mitochondrial gene codons in five *Japanagallia* species. Meanwhile, our results showed that AT and GC are not used equally frequently at the third codon in 13 mitochondrial genes, suggesting that mutational pressure and natural selection may affect the codon preference in these genes.

## Analysis the relationship of ENC and compositional attributes

The average ENC value for mitochondrial genes of the *Japanagallia* in our analysis was 34.564 (*ATP6*), 39.0844 (*ATP8*), 38.3864 (*COX1*), 36.4992 (*COX2*), 37.3108 (*COX3*),

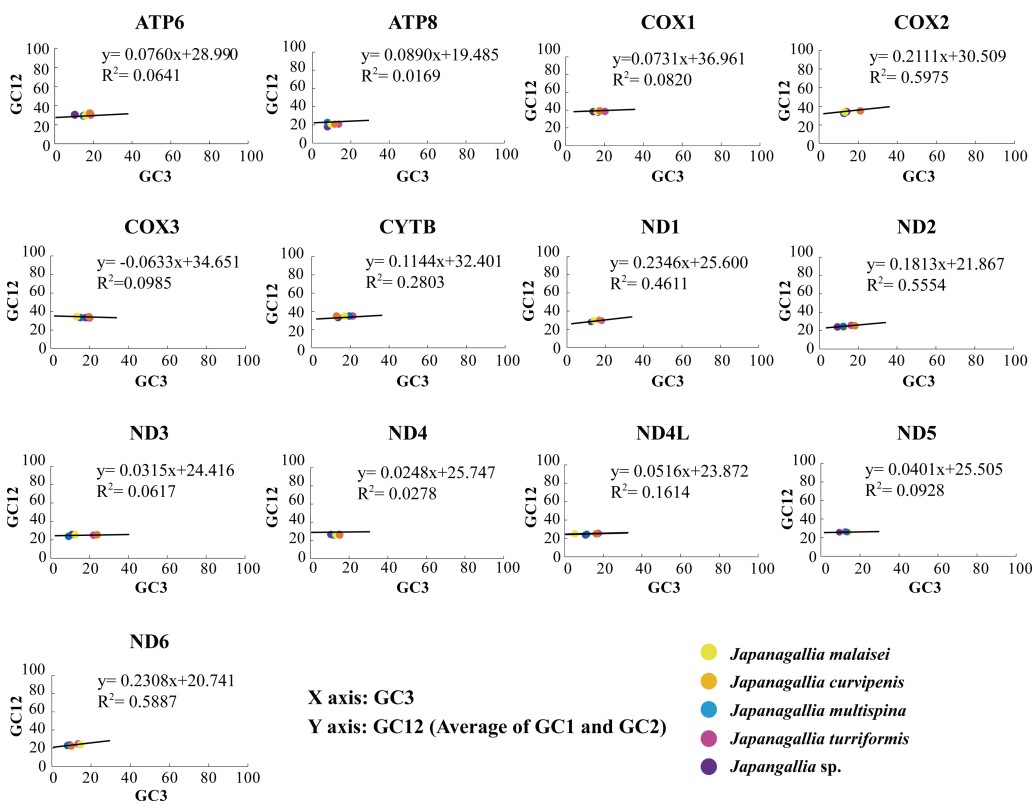

**Figure 5** Neutrality plot analysis of 13 mitochondrial protein-coding genes in five *Japanagallia* species.

36.1256 (*CYTB*), 31.1794 (*ND1*), 36.1856 (*ND2*), 34.291 (*ND3*), 32.482 (*ND4*), 31.2502 (*ND4L*), 33.8502 (*ND5*) and 32.1766 (*ND6*), which represented relatively high codon bias in these genes. At the same time, in order to better understand the impact of translational selection or mutation pressure on the codon usage of mitochondrial genes of the *Japanagallia*, we performed correlation analysis between ENC, the overall composition of nucleotide and its third position of the codon (Table 3). We found that in some mitochondrial genes, there was a significant positive correlation between homogenous nucleotides, while there was a significant negative correlation between most heterogeneous nucleotides, suggesting that the base composition bias of *Japanagallia* was affected by mutation pressure (*Wong et al., 2010*). In addition, there was a significant positive correlation between the ENC of *COX2*, *ND3* and *ND4* genes and GC3 ($P < 0.05$), indicating that the codon preference of these genes may be affected by GC content.

## Correspondence analysis of codon bias

In order to investigate the trends of codon usage variation in five *Japanagallia* species for the 13 mitochondrial protein-coding genes, we performed COA using the RSCU value of codons. The distribution of axes in 13 mitochondrial genes is shown in Fig. 7. In our analysis, all zero row codons and two stop codons (UAA, UAG) in the matrix were deleted. We observed that the first principal axis accounted for 33.63% (*ATP6*), 47.67% (*ATP8*),

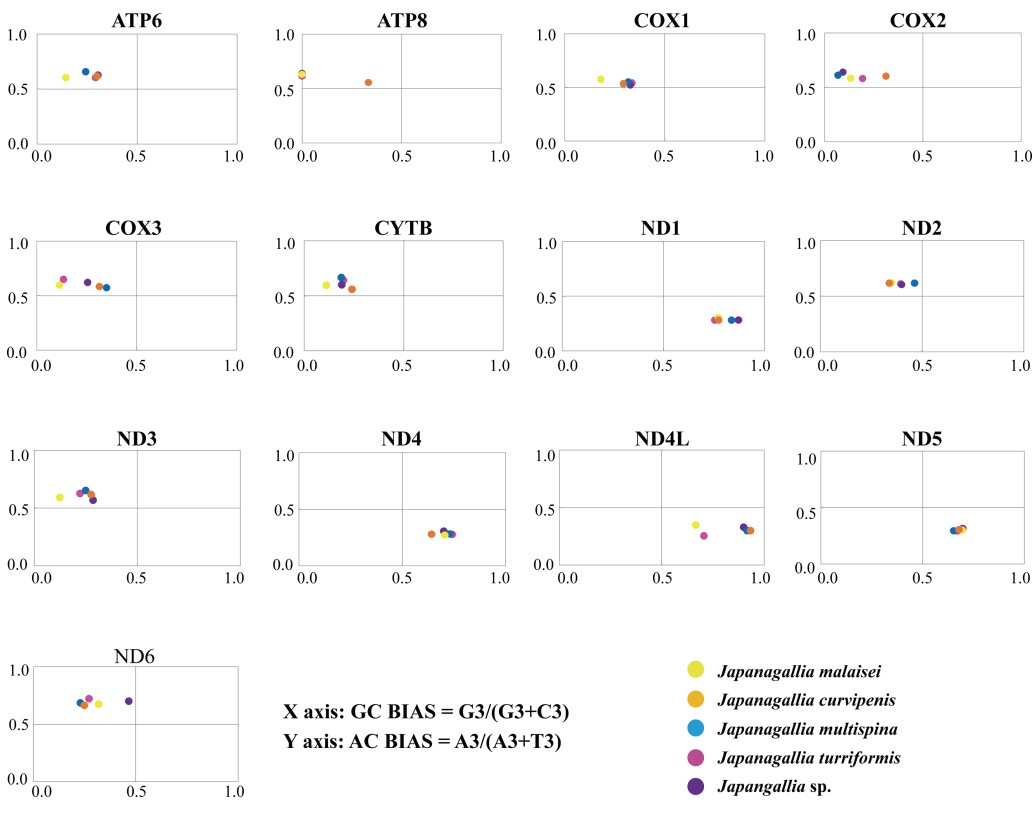

**Figure 6** Parity plot analysis of 13 mitochondrial protein-coding genes in five *Japanagallia* species.

**Table 2** The average percentage of GC bias and AT bias in the CDS of 13 mitochondrial protein-coding genes of five *Japanagallia* species.

| Sl. No | Genes | GC BIAS G3/(G3 + C3) | AT BIAS A3/(A3 + T3) |
|---|---|---|---|
| 1 | ATP6 | 0.26 | 0.62 |
| 2 | ATP8 | 0.67 | 0.62 |
| 3 | COX1 | 0.29 | 0.54 |
| 4 | COX2 | 0.16 | 0.60 |
| 5 | COX3 | 0.23 | 0.60 |
| 6 | CYTB | 0.18 | 0.61 |
| 7 | ND1 | 0.80 | 0.29 |
| 8 | ND2 | 0.39 | 0.61 |
| 9 | ND3 | 0.23 | 0.61 |
| 10 | ND4 | 0.70 | 0.28 |
| 11 | ND4L | 0.82 | 0.31 |
| 12 | ND5 | 0.68 | 0.30 |
| 13 | ND6 | 0.31 | 0.69 |

**Table 3** Correlation analysis between ENC, the overall composition of nucleotide and its third position of the codon.

| ATP6 | A3% | T3% | G3% | C3% | GC3% |
|---|---|---|---|---|---|
| A% | 0.985** | −0.179 | −0.445 | −0.850 | −0.867 |
| T% | 0.002 | 0.639 | −0.800 | −0.005 | −0.349 |
| G% | −0.352 | −0.101 | 0.918* | 0.004 | 0.400 |
| C% | −0.870 | −0.196 | 0.607 | 0.877 | 0.957* |
| GC% | −0.816 | −0.191 | 0.792 | 0.706 | 0.902* |
| ENC | −0.782 | 0.016 | 0.755 | 0.542 | 0.756 |
| **ATP8** | | | | | |
| A% | 0.911* | −0.583 | −0.583 | NO | −0.915* |
| T% | −0.716 | 0.612 | 0.612 | NO | 0.560 |
| G% | −0.991** | 0.772 | 0.772 | NO | 0.853 |
| C% | 0.012 | −0.232 | −0.232 | NO | 0.219 |
| GC% | −0.541 | 0.238 | 0.238 | NO | 0.654 |
| ENC | −0.809 | 0.992** | 0.992** | NO | 0.325 |
| **COX1** | | | | | |
| A% | 0.930* | −0.139 | −0.804 | −0.270 | −0.704 |
| T% | −0.274 | 0.917* | 0.196 | −0.935* | −0.563 |
| G% | −0.906* | −0.023 | 0.942* | 0.318 | 0.825 |
| C% | −0.222 | −0.618 | 0.119 | 0.907* | 0.741 |
| GC% | −0.593 | −0.447 | 0.573 | 0.794 | 0.920* |
| ENC | −0.416 | −0.361 | 0.167 | 0.553 | 0.511 |
| **COX2** | | | | | |
| A% | 0.868 | 0.139 | −0.927* | −0.522 | −0.824 |
| T% | 0.142 | 0.918* | −0.598 | −0.977** | −0.788 |
| G% | −0.933* | −0.177 | −0.981** | 0.638 | 0.906* |
| C% | −0.280 | −0.790 | 0.708 | 0.850 | 0.810 |
| GC% | −0.721 | −0.509 | 0.967** | 0.825 | 0.971* |
| ENC | −0.929* | −0.177 | 0.964** | 0.658 | 0.902* |
| **COX3** | | | | | |
| A% | 0.970** | −0.304 | −0.940 | 0.259 | −0.393 |
| T% | −0.628 | 0.900* | 0.523 | −0.950* | −0.551 |
| G% | −0.889* | 0.206 | 0.933** | −0.244 | 0.443 |
| C% | 0.408 | −0.792 | −0.421 | 0.923* | 0.596 |
| GC% | −0.335 | −0.576 | 0.407 | 0.669 | 0.916* |
| ENC | −0.493 | 0.194 | 0.510 | −0.202 | 0.154 |
| **CYTB** | | | | | |
| A% | 0.980** | −0.580 | 0.229 | 0.204 | 0.237 |
| T% | −0.680 | 0.992** | −0.625 | −0.881* | −0.926* |
| G% | −0.293 | −0.310 | 0.760 | 0.405 | 0.554 |
| C% | 0.304 | −0.876 | 0.511 | 0.971* | 0.972* |
| GC% | 0.200 | −0.824 | 0.604 | 0.926* | 0.958* |
| ENC | −0.083 | −0.570 | 0.662 | 0.698 | 0.779 |

 

**Table 3** (*continued*)

| ATP6 | A3% | T3% | G3% | C3% | GC3% |
|------|------|------|------|------|------|
| ND1 | | | | | |
| A% | 0.860 | 0.678 | −0.972* | −0.806 | −0.935* |
| T% | 0.327 | 0.933* | −0.700 | −0.930* | −0.844 |
| G% | −0.516 | −0.960* | 0.857 | 0.991* | 0.961* |
| C% | −0.543 | −0.609 | 0.700 | 0.679 | 0.721 |
| GC% | −0.587 | −0.887* | 0.872 | 0.941* | 0.945* |
| ENC | −0.069 | −0.495 | 0.296 | 0.469 | 0.394 |
| ND2 | | | | | |
| A% | 0.967* | 0.769 | −0.720 | −0.906* | −0.905* |
| T% | 0.800 | 0.990* | −0.774 | −0.899* | −0.918* |
| G% | −0.800 | −0.689 | 0.785 | 0.694 | 0.774 |
| C% | −0.956* | −0.891* | 0.693 | 0.991* | 0.957* |
| GC% | −0.982* | −0.894* | 0.788 | 0.967* | 0.973* |
| ENC | −0.866 | −0.751 | 0.793 | 0.782 | 0.840 |
| ND3 | | | | | |
| A% | 0.964** | 0.247 | −0.624 | −0.845 | −0.816 |
| T% | −0.045 | 0.799 | −0.626 | −0.369 | −0.465 |
| G% | −0.879* | −0.497 | 0.840 | 0.886* | 0.912* |
| C% | −0.893* | −0.580 | 0.840 | 0.699* | 0.974** |
| GC% | −0.906* | −0.567 | 0.857 | 0.963** | 0.974* |
| ENC | −0.835 | −0.643 | 0.781 | 0.993** | 0.973** |
| ND4 | | | | | |
| A% | 0.896* | −0.241 | −0.769 | −0.221 | −0.630 |
| T% | 0.096 | 0.916* | −0.411 | −0.601 | −0.565 |
| G% | −0.757 | −0.381 | 0.981** | 0.360 | 0.841 |
| C% | −0.571 | −0.773 | 0.666 | 0.922** | 0.890* |
| GC% | −0.747 | −0.627 | 0.930* | 0.693 | 0.963** |
| ENC | −0.730 | −0.571 | 0.960** | 0.555 | 0.919* |
| ND4L | | | | | |
| A% | 0.958* | 0.151 | −0.853 | −0.398 | −0.936 |
| T% | 0.384 | 0.606 | −0.227 | −0.567 | −0.476 |
| G% | −0.790 | −0.664 | 0.971** | 0.021 | 0.895* |
| C% | −0.395 | 0.623 | −0.159 | 0.950* | 0.230 |
| GC% | −0.968** | −0.296 | 0.841 | 0.534 | 0.978** |
| ENC | −0.391 | 0.098 | 0.116 | 0.609 | 0.346 |
| ND5 | | | | | |
| A% | 0.912* | 0.247 | −0.629 | −0.927* | −0.761 |
| T% | 0.512 | 0.793 | −0.868 | −0.516 | −0.764 |
| G% | −0.840 | −0.794 | 0.986** | 0.917* | 0.992** |
| C% | −0.946* | −0.615 | 0.935* | 0.959** | 0.974** |
| GC% | −0.889 | −0.187 | 0.632 | 0.796 | 0.714 |
| ENC | −0.606 | −0.851 | 0.936* | 0.649 | 0.859 |

**Table 3** (*continued*)

| ATP6 | A3% | T3% | G3% | C3% | GC3% |
|------|-----|-----|-----|-----|------|
| ND6 | | | | | |
| A% | 0.853 | −0.297 | 0.005 | −0.654 | −0.507 |
| T% | 0.102 | 0.888[*] | −0.743 | −0.647 | −0.818 |
| G% | 0.266 | −0.850 | 0.542 | 0.300 | 0.463 |
| C% | −0.676 | −0.401 | 0.552 | 0.888[*] | 0.924[*] |
| GC% | −0.408 | −0.741 | 0.709 | 0.857 | 0.967[**] |
| ENC | −0.291 | −0.717 | 0.313 | 0.914[*] | 0.844 |

**Notes.**
[**]$p < 0.01$.
[*]$p < 0.05$.

36.49% (*COX1*), 32.93% (*COX2*), 35.40% (*COX3*), 39.00% (*CYTB*), 41.17% (*ND1*), 33.9% (*ND2*), 36.09% (*ND3*), 36.56% (*ND4*), 43.15% (*ND4L*), 39.16% (*ND5*), 38.02% (*ND6*) of all variations in the gene set. The second axis accounted for 28.99% (*ATP6*), 40.47% (*ATP8*), 28.15% (*COX1*), 27.86% (*COX2*), 24.81% (*COX3*), 24.28% (*CYTB*), 24.10% (*ND1*), 30.60% (*ND2*), 27.66% (*ND3*), 29.96% (*ND4*), 33.69% (*ND4L*), 27.82% (*ND5*), 33.62% (*ND6*) of all variations within the gene get.

In addition, by observation we found that codons were mostly located near the axes and concentrated in the center of the graph, indicating that codon bias of these genes may be related to the base composition of mutation bias supporting the view of *Butt, Nasrullah & Tong (2014)*. However, it was also observed that some gene codons were discretely distributed, indicating that the codon usage of 13 mitochondrial genes might be affected by natural selection and other factors (*Wei et al., 2014*).

## Grand average of hydropathy analysis

We calculated the protein properties and amino acid composition of 13 protein-coding genes in the mitochondrial genome of five *Japanagallia* species. The grand average of hydrophilicity (GRAVY) of the 13 mitochondrial protein-coding genes (Table 4) was calculated to be positive, suggesting that the proteins of the *Japanagallia* are hydrophobic and may be to maintain their biological functions. At the same time, the overall frequency of the usage amino acid in the mitochondrial genes (Fig. 8) showed that the contents of Leucine (Leu), Isoleucine (Ile), Methionine (Met), Serine (Ser) and Phenylalanine (Phe) were higher than those of other amino acids, and Arginine (Arg) had the lowest content.

## Phylogenetic analysis

The phylogenetic relationships were analyzed using ML and BI to reconstruct among the 81 species of the 12 subfamilies of leafhoppers, five species of treehoppers, and two outgroup species based on the following three datasets (AA, PCGRNA, PCG12RNA), and six phylogenetic trees were obtained (ML-AA, BI-AA, ML-PCGRNA, BI-PCGRNA, ML-PCG12RNA and BI-PCG12RNA) (Figs. 9 and 10; Figs. S1–S5).

In the six trees, most nodes received high nodal support values, and a few nodes only obtain medium or low support, and the monophyly of each subfamily was generally well supported, while the relationship among subfamilies were discrepant across analyses. At the same time, all phylogenetic trees showed the treehoppers clustered into one group and

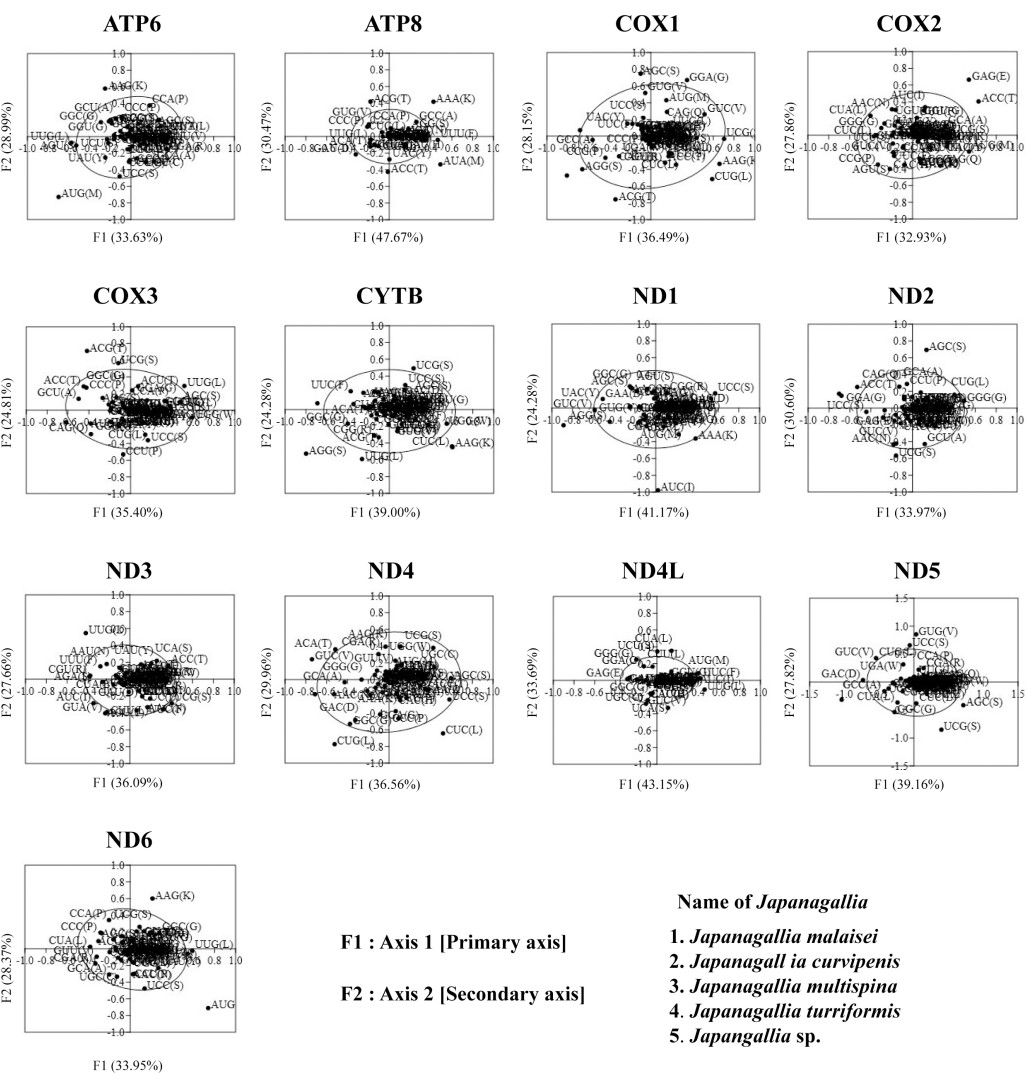

**Figure 7 Correspondence analysis of the relative synonymous codon usage values of 13 mitochondrial protein-coding genes in five *Japanagallia* species.** The black color dot indicates codons encoding amino acids of the genes.

a sister group of Megophthalminae with high support, which supported the previous view that leafhoppers originated from the Cicadellinae, and further confirm that Cicadellidae is a paraphyletic group. These results are consistent with many previous studies based on mitogenomes (*Dietrich et al., 2001*, *Dietrich et al., 2017*; *Zhao & Liang, 2016*; *Du, Dai & Dietrich, 2017*; *Du et al., 2017*; *Du, Dietrich & Dai, 2019*; *Skinner et al., 2020*; *Jiang et al., 2021*; *Wang, Wang & Dai, 2021*). Within Megophthalminae, all phylogenetic trees showed that *Japanagallia* and *Durgade* were sister groups, which was consistent with previous reports based on molecular data (*Wang et al., 2017*).

## DISCUSSION

In this study, the complete mitochondrial genomes of five *Japanagallia* species were studied. the mitochondrial genome sizes of five species ranged from 15,365 bp (*J. curvipenis*) to 15,717 bp (*J. turriformis*), and the differences were mainly influenced by the length of CRs. The comparative analyses showed that the mitochondrial genome structure and composition of five species were very similar to those of the published leafhoppers. At the same time, we analyzed the base composition of codons and found that the overall percentage of nucleotide and their composition at the third codon position of the 13 protein-coding genes of the five *Japanagallia* species were A >T >C >G and T3 >A3 >C3 >G3, respectively. The GC content was generally lower than the AT content, that is, the genes were rich in AT. This result supports previous studies on the mitochondrial genome of *Atkinsoniella zizhongi* (*Jiang et al., 2022*), *Centrotoscelus davidi* (*Li et al., 2022*), *Bhatia longiradiata* (*Lu, Huang & Deng, 2023*). In addition, AT richness was also reported in mitochondrial genomes of soft scales (*Lu, Huang & Deng, 2023*) supporting this study.

The codon usage patterns are mainly influenced by mutation pressure and natural selection (*Sharp, Emery & Zeng, 2010*), while base composition (*Li et al., 2022*), protein hydrophobicity and gene length (*Prat et al., 2009*) also have certain effects. In this study, the results of RSCU analysis showed that codons ending in A/T bases were used more frequently and codons ending in G/C bases were used less frequently in the mitochondrial genomes of five *Japanagallia* species, which was consistent with the analysis of *Atkinsoniella* (*Jiang et al., 2021*), *Cladolidia* (*Wang, Wang & Dai, 2021*), *Eupteryx minusula* and *Eupteryx gracilirama* (*Yuan et al., 2021*). The neutrality plot showed that the correlation between GC12 and GC3 was weak, and the regression slope was close to zero, it revealed that the codon usage pattern was mainly affected by natural selection. At the same time, combined with analysis of PR2-plot, ENC-plot, COA, and GRAVY, we found that codon usage patterns were influenced by many factors, but natural selection was the dominant factor supporting the previous research work in the pea aphid genome (*Rispe et al., 2007*), *Bombyx mori* mitogenome (*Abdoli et al., 2022*). Although we do not know the cause of this phenomenon now, there should be a balance between natural selection and mutation pressure in biological growth to be further explored (*Wang, Meng & Wei, 2018*; *Lu, Huang & Deng, 2023*).

We generated BI and ML trees based on three datasets. Our results indicated that each subfamily was separated into monophyletic groups, and some relationships are highly stable and supported. For example, Iassinae emerged as the sister group to Coelidiinae; Mileewinae emerged as a sister group with Evacanthine and Ledrinae in the six phylogenetic trees (Fig. 9), which is consistent with the results of previous studies (*Wang et al., 2018*; *Wang et al., 2019*; *Wang et al., 2020a*; *Wang et al., 2020b*; *Jiang et al., 2021*; *Wang, Wang & Dai, 2021*; *He et al., 2022*; *Lu et al., 2022*). However, this finding is different from the results of *Du, Dietrich & Dai (2019)*, *Wang et al. (2020a)*, *Wang et al. (2020b)* and *Wang, Wang & Dai (2021)*. Their phylogenetic analysis showed that Megophthalminae is a sister group of Macropsinae rather than treehoppers, which may be due to the different datasets used. Seven species from two genera (*Japanagallia*: *J. multispina*, *J. turriformis*, *J. spinsa*, *J.*

Li et al. (2023), *PeerJ*, DOI 10.7717/peerj.16058

**Table 4  Grand average of hydropathy analysis of 13 mitochondrial protein-coding genes in five *Japanagllia* species.**

| *Japanagallia* | ATP6 | ATP8 | COX1 | COX2 | COX3 | CYTB | ND1 | ND2 | ND3 | ND4 | ND4L | ND5 | ND6 |
|---|---|---|---|---|---|---|---|---|---|---|---|---|---|
| *Japanagallia malaisei* | 0.842 | 0.139 | 0.637 | 0.173 | 0.429 | 0.429 | 0.985 | 0.893 | 1.021 | 1.224 | 1.577 | 1.007 | 0.732 |
| *Japanagallia curvipenis* | 0.788 | 0.047 | 0.648 | 0.234 | 0.478 | 0.603 | 0.931 | 0.887 | 1.021 | 1.271 | 1.597 | 1.011 | 0.821 |
| *Japanagallia multispina* | 0.781 | 0.139 | 0.665 | 0.204 | 0.459 | 0.544 | 0.943 | 0.824 | 0.978 | 1.232 | 1.604 | 1.068 | 0.766 |
| *Japanagallia turriformis* | 0.864 | 0.339 | 0.656 | 0.189 | 0.472 | 0.562 | 0.981 | 0.888 | 1.162 | 1.252 | 1.584 | 1.028 | 0.726 |
| *Japanagallia* sp. | 0.737 | 0.157 | 0.648 | 0.25 | 0.437 | 0.537 | 0.911 | 0.845 | 1.05 | 1.249 | 1.519 | 1.046 | 0.657 |
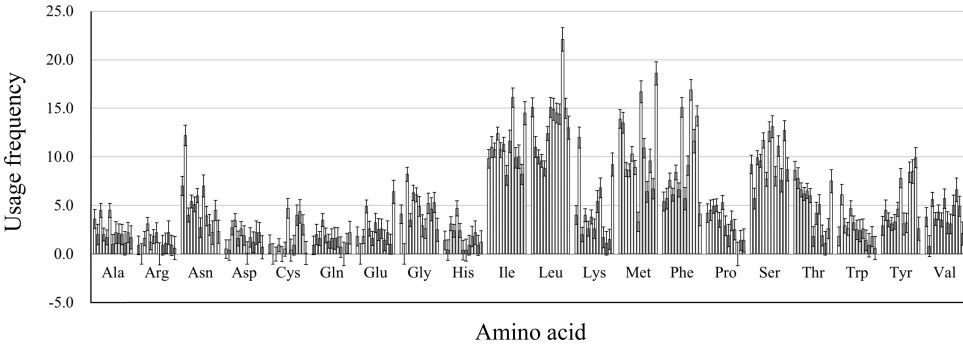

**Figure 8** Amino acid usage frequency for 13 mitochondrial protein-coding genes in five *Japanagallia* species.

*malaise*, *J. curvipenis*, *Japanagallia* sp.; *Durgades*: *D. nigropicta*) of the Megophthalminae subfamily were analyzed. All phylogenetic analyses showed that Megophthalminae is located in the middle of the phylogenetic tree with high support values (BS = 100, PP = 1), which was similar to the results of *He et al. (2022)* and *Wang et al. (2022)*. At the same time, the monophyly of Megophthalminae was well supported in our study. This result was consistent with the previous phylogenetic analysis based on anchored hybrid enrichment genomics, mitochondrial genome and transcriptome data (*Dietrich et al., 2017*; *Wang et al., 2017*; *Hu et al., 2022*).

Within *Japanagallia*, based on the six phylogenetic trees, we found that six *Japanagallia* species clustered into a monophyletic group and formed a sister group with *Durgade*, and all species maintained the same relationships and topologies. The internal genetic relationship of the subfamily Megophthalminae were as follows: (*D. nigropicta* + (*J. multispina* + (*J. turriformis* + ((*J. spinsa* + *J. malaisei*)) + (*Japanagallia* sp. + *J. curvipenis*)))) (Fig. 10 and Figs. S1–S3). The phylogenetic relationships with strong support values in BI (PP > 0.97) and low to high support values in ML (BS = 59–100). In our analysis, only used six *Japanagallia* species and one *Durgades* species, while other genera and species were not involved, it is not sufficient to indicate the phylogenetic relationship and clarify the monophyletic problem between the genera *Japanagallia* and *Durgades*. Our study has elucidated the relationships within the genus *Japanagallia*, and it is necessary to obtain more new sequencing data in the future to enrich the mitochondrial genome database of Megophthalminae, which will help us better reveal the evolution and phylogenetic relationships of Megophthalminae.

## CONCLUSION

In our study, complete mitogenomes of five *Japanagallia* species were newly sequenced, and the codon usage patterns and phylogenetic relationships were compared and analyzed. Their length, composition and structure were consistent with previous studies of leafhoppers. The results of codon usage bias indicated that codons ending with A/T were naturally preferred among the 13 protein coding genes in the mitogenomes of *Japanagallia* species.

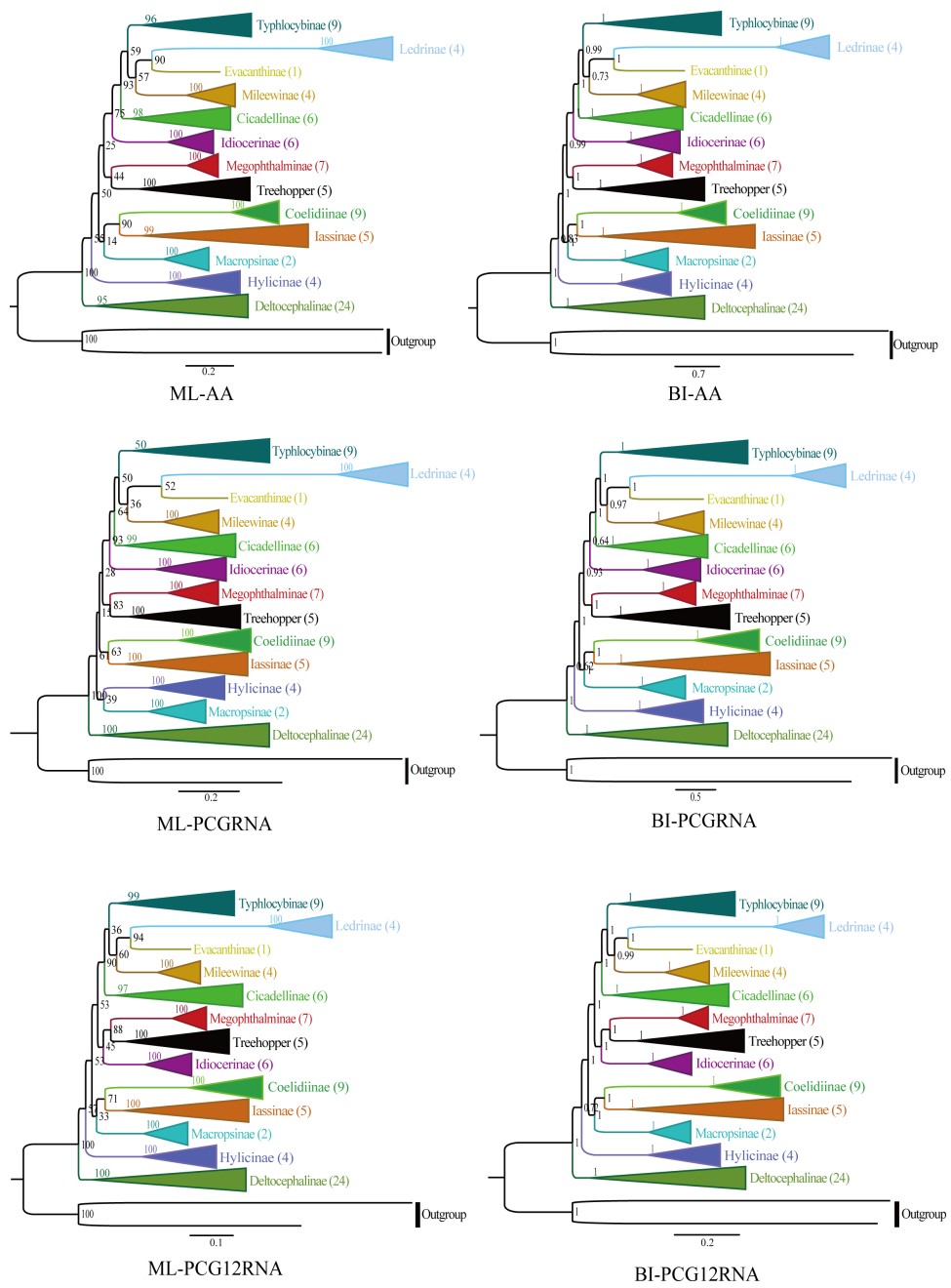

**Figure 9** Phylogenetic trees of leafhoppers were inferred from different mitochondrial genome datasets by maximum likelihood (ML) and Bayesian analyses (BI) methods.

By comparing the genomes of five *Japanagallia* species and their AT-GC bias showed the effects of AT bias in shaping the pattern of codon usage. Furthermore, the results of ENC plot, PR2 plot, neutrality plot and COA analysis revealed that natural selection played a dominant role in the codon usage of mitochondrial genes in five *Japanagallia* species, while mutation pressure had little effect. Phylogenetic analysis is based on three datasets (AA,

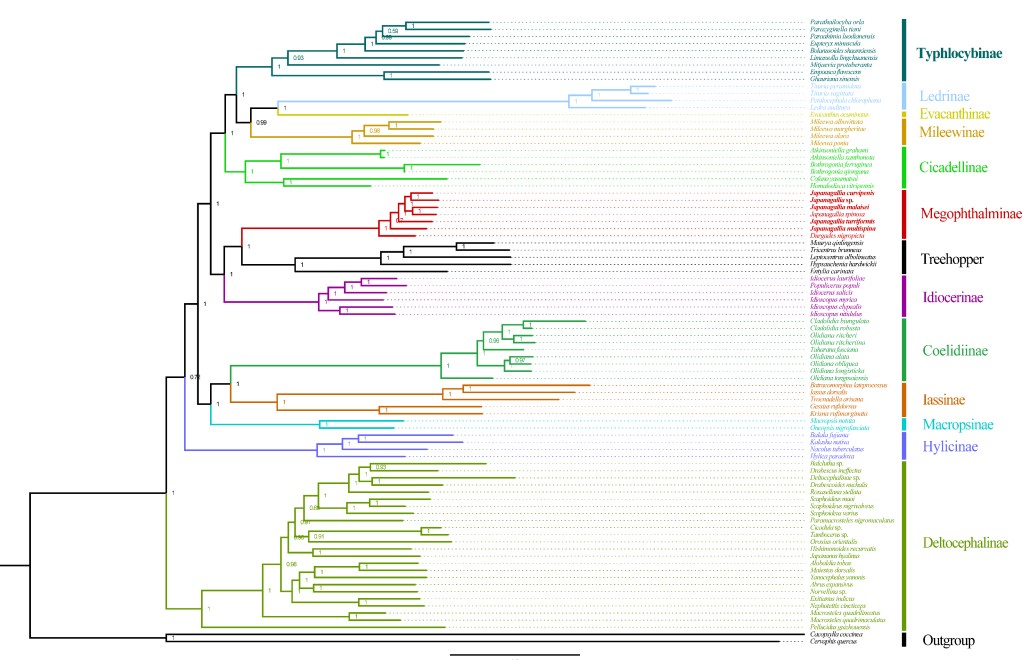

**Figure 10** Phylogeny tree constructed using BI method to the 1st and 2nd codon locations of 13 PCGs and two rRNA genes.

PCGRNA, PCG12RNA) with two methods (ML and BI), and supported the monophyly of Megophthalminae with high support values. In addition, the phylogenetic relationship among the species of *Japanagallia* was confirmed. Our results will not only provide a new understanding of the importance of the codon usage patterns of the 13 protein-coding genes in the mitogenomes of *Japanagallia* and provide basic data for its biological evolution and phylogeny, but also help to understand the codon usage patterns of the mitogenomes of other leafhoppers. Meanwhile, more mitogenomic data are needed to better verify the monophyly and phylogenetic relationships of Megophthalminae in the future.

## ACKNOWLEDGEMENTS

We are very grateful to Chao Zhang (Institute of Entomology, Guizhou University, Guiyang, China) for providing material for this study, and to Jikai Lu, Kai Yu, Qin Yang, Die Liu and Meishu Guo for reading the original manuscript.

### Funding

This work was supported by the National Natural Science Foundation of China (No. 32160119); the National Natural Science Foundation of China (No. 32000329); the Program of Excellent Innovation Talents, Guizhou Province, China (No. 20206003); and the Opening Foundation of Shaanxi University of Technology (No. SLGPT2019KF03-01).

The funders had no role in study design, data collection and analysis, decision to publish, or preparation of the manuscript.

### Grant Disclosures

The following grant information was disclosed by the authors:

The National Natural Science Foundation of China: No. 32160119.

The National Natural Science Foundation of China: No. 32000329.

The Program of Excellent Innovation Talents, Guizhou Province, China: No. 20206003.

Opening Foundation of Shaanxi University of Technology: SLGPT2019KF03-01.

### Competing Interests

The authors declare there are no competing interests.

### Author Contributions

- Min Li conceived and designed the experiments, analyzed the data, prepared figures and/or tables, authored or reviewed drafts of the article, and approved the final draft.
- Jiajia Wang performed the experiments, analyzed the data, authored or reviewed drafts of the article, and approved the final draft.
- Renhuai Dai conceived and designed the experiments, performed the experiments, authored or reviewed drafts of the article, and approved the final draft.
- Guy Smagghe conceived and designed the experiments, authored or reviewed drafts of the article, and approved the final draft.
- Xianyi Wang conceived and designed the experiments, authored or reviewed drafts of the article, and approved the final draft.
- Siying You conceived and designed the experiments, authored or reviewed drafts of the article, and approved the final draft.

### Data Availability

The sequence data are available at GenBank: *Japanagallia curvipenis* (OQ612705), *Japanagallia malaisei* (OQ658733), *Japanagallia multispina* (OQ630909), *Japanagallia turriformis* (OQ646650), and *Japanagallia* sp (OQ654105).

### Supplemental Information

Supplemental information for this article can be found online at http://dx.doi.org/10.7717/peerj.16058#supplemental-information.

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
