# Peer review of "Comparative analysis of codon usage patterns and phylogenetic implications of five mitochondrial genomes of the genus Japanagallia Ishihara, 1955 (Hemiptera, Cicadellidae, Megophthalminae)"

_PeerJ, doi:10.7717/peerj.16058_

## Round 0.1 · original submission · Major Revisions

Dear Dr. Li and colleagues:

Thanks for submitting your manuscript to PeerJ. I have now received two independent reviews of your work, and as you will see, the reviewers raised some concerns about the research (mostly the manuscript format and content). Despite this, these reviewers are optimistic about your work and the potential impact it will have on research studying cicadellid systematics and mitochondrial evolution. Thus, I encourage you to revise your manuscript, accordingly, taking into account all of the concerns raised by both reviewers.

Focus on clarity, and address the sections considered incomplete and/or unclear by the reviewers. It appears that certain key references are missing. The Methods should be clear, concise and repeatable. Please ensure this. Also, elaborate on the discussion of your findings, placing them within a broad and inclusive body of work by the field. Spend some time fixing the figures as recommended by the reviewers.

While the concerns of the reviewers are relatively minor, this is a major revision to ensure that the original reviewers have a chance to evaluate your responses to their concerns. There are not too many suggestions; thus, it should not take much effort to address these concerns to greatly improve your manuscript.

I look forward to seeing your revision, and thanks again for submitting your work to PeerJ.

Good luck with your revision,

-joe

Reviewer 1 ·

Basic reporting

no comment

Experimental design

no comment

Validity of the findings

no comment

Additional comments

Line 39: it seems there is an extra . after “hydrophobicity”.

Line 102-104:Geneious Primer was used to assemble mitogenome using NC_065685, with this method, it is easy to detect degenerate bases in the obtained assembled sequences, how authors dealt with this situation. In addition, please supplement how the authors confirmed the control region is complete.

Line 105: Change to “The assembled mitogenomes were annotated using MITOS web server with the invertebrate genetic code.”

Line 123: delete “using” or “by”.

Line 199: The genus of J. sp. should be written completely.

Line 206: change “GC3“ to “C3”.

Line 348-350: it is difficult to quickly read the relationships among subfamilies, another way to express is needed.

Line 351: The genus of J. sp. should be written completely.

Line 368-370: This study can indeed provide a reference for the relationship between the subfamily Megophthalminae and other subfamilies and further enrich the mitochondrial genome database, while “Therefore, it is necessary to add more mitochondrial genome data of Megophthalminae in future studies to elucidate the phylogenetic relationship.” is not the consequence of last sentence.

Line 388: “however” always used to introduce a statement that contrasts with or seems to contradict something that has been said previously, so, it not propriate use in this sentence.

The title of Figure 2: there is an extra space in “t hird”.

Figure 4: Cluster analysis is recommended to better show the differences of the relative synonymous codon usage values in each gene.

Figure 5 and 6: plot of each species using different color could be better.

Figure 7: In the bottom right corner of the figure, 1-5 were used to represent five Japanagallia species, but in the figure of each gene, 1-5 cannot be found. I suggest using different colors to represent different species in each gene and note correctly.

Reviewer 2 ·

Basic reporting

This article newly sequenced and assembled five mitochondrial genomes of Japanagallia. The codon usage bias, nucleotide composition and relative synonymous codon usage are analyzed in detail. The phylogenetic relationships of Cicadellidae were reconstructed using maximum likelihood method and Bayesian method combined with 73 mitochondrial genomes of Cicadellidae from public data. The methods are detailed and logical, but there are a lot of improvements that need to be made before publication.
The introduction needs to be more refined, highlighting what problems are being addressed, rather than devoting a lot of space to the mitochondrial genome -- which everyone already knows. Lines 54 -- 70 need to be rewritten, in fact species identification is irrelevant to this article. Since you reconstructed the phylogenetic analysis, why not write in the introduction what the phylogeny of your taxa looks like now? What unresolved issues remain?
In the discussion, it seems unreasonable for the author to compare the results with Bombyx mori, fishes, birds and mammals. Hemiptera have a large number of mitochondrial genome studies, so why not compare them with these more closely related species.

Experimental design

The authors have sequenced, spliced and annotated the complete mitochondrial genome, but why did not concatenate tRNA genes when constructing the phylogenetic tree? Because of the heterogeneity of mitochondrial genomes, phylogenetic trees can be constructed by heterogeneous models such as phylobayes. In addition, the article uses a large amount of public data, which requires the source. Moreover, nuclear genes can also be extracted from SRA data and then concatenate with mitochondrial genomes for phylogenetic analysis, the results may be more convincing.

Validity of the findings

This article newly sequenced and assembled five mitochondrial genomes of Japanagallia. The codon usage bias, nucleotide composition and relative synonymous codon usage are analyzed in detail. The phylogenetic relationships of Cicadellidae were reconstructed using maximum likelihood method and Bayesian method combined with 73 mitochondrial genomes of Cicadellidae from public data.

Additional comments

There is no need to make Fig2 a 3D image, in fact, a 2D bar image would look better. Fig10 resolution is too low and needs to be improved.

---

## Round 0.2 · Minor Revisions

Dear Dr. Li and colleagues:

Thanks for revising your manuscript. The reviewers are very satisfied with your revision (as am I). Great! However, there are a few minor issues to attend to. Please address these ASAP so we may move towards acceptance of your work.

Best,

-joe

Reviewer 1 ·

Basic reporting

The authors have answered all the questions I proposed, but there are still some issues need to be corrected. The line numbers are

Line 267 (in the PDF file): The “p“ in the bracket should be capitalized and italicized.

Line 323: delete “sp. nov.”, or change it to “Jiang & Yang, 2022”.

Line 348: change “(Du et al., 2019) and (Wang et al., 2020, 2021)” to “Du et al. (2019) and Wang et al. (2020, 2021)”.

Line 388: change “datas” to “data”.

The “J. sp.” in table S2 should be completely written as “Japanagallia sp.”.

The paragraph below the title should be placed as a note below the whole table.

Figure 2: It will be better with a white background.

The *.gb file of each sequence (in the peerj-85040-Five_sequence_information.zip folder): In the manuscript, these five mitogenomes are circular, but in all the gb files, they are noted as linear, please correct. And the accession and organism in each file should be supplemented.

Experimental design

no comment

Validity of the findings

no comment

---

## Round 0.3 · accepted · Accept

Dear Dr. Li and colleagues:

Thanks for revising your manuscript based on the concerns raised by the reviewers. I now believe that your manuscript is suitable for publication. Congratulations! I look forward to seeing this work in print, and I anticipate it being an important resource for research studying cicadellid systematics and mitochondrial evolution. Thanks again for choosing PeerJ to publish such important work.

Best,

-joe